# The Impact of Profiles Data Assimilation on an Ideal Tropical Cyclone Case

Changliang Shao [1,2,*] and Lars Nerger [2]

1   China Meteorological Administrator Meteorological Observation Centre, Beijing 100081, China
2   Alfred-Wegener-Institute, Helmholtz-Zentrum für Polar-und Meeresforschung (AWI),
    27570 Bremerhaven, Germany; lars.nerger@awi.de
*   Correspondence: shaocl@cma.gov.cn

**Abstract:** Profile measurements play a crucial role in operational weather forecasting across diverse scales and latitudes. However, assimilating tropospheric wind and temperature profiles remains a challenging endeavor. This study assesses the influence of profile measurements on numerical weather prediction (NWP) using the weather research and forecasting (WRF) model coupled to the parallel data assimilation framework (PDAF) system. Utilizing the local error-subspace transform Kalman filter (LESTKF), observational temperature and wind profiles generated by WRF are assimilated into an idealized tropical cyclone. The coupled WRF-PDAF system is adopted to carry out the twin experiments, which employ varying profile densities and localization distances. The results reveal that high-resolution observations yield significant forecast improvements compared to coarser-resolution data. A cost-effective balance between observation density and benefit is further explored through the idealized tropical cyclone case. According to diminishing marginal utility and increasing marginal costs, the optimal observation densities for U and V are found around 26–27%. This may be useful information to the meteorological agencies and researchers.

**Keywords:** data assimilation; LESTKF; profile; density; localization distance

## 1. Introduction

Profile observations have gained considerable attention in recent years for their potential to enhance the performances of atmospheric models and weather forecasting for the deeply developed weather systems, e.g., the tropical cyclones. These profiles can be derived from various remote-sensing instruments, including radiosondes, dropsondes, wind profiler radar, ground and space-based lidars, and microwave radiometers. Assimilating profiles into atmospheric models using techniques like ensemble Kalman filtering (EnKF) has shown a significant impact on improving the accuracy and performance of global models [1], mesoscale models [2], and even small-scale models [3]; however, the solution for mitigating typhoon disasters through the use of these models, considering the observation cost and forecast benefit, is still under-explored. Therefore, the cost accounting of profile observations used to improve numerical forecasting is of great significance for both practical numerical weather forecasting and observation network construction.

Data assimilation (DA) plays a pivotal role in enhancing the accuracy and reliability of numerical weather prediction (NWP) models. It serves as the bridge between model simulations and real-world observations, elevating the accuracy, skill, and reliability of atmospheric simulations. DA has wide-ranging applications, including weather forecasting, climate studies, and environmental assessments [4,5]. NWP model outputs have witnessed continuous improvements, owing in part to advancements such as more precise initial states, an increased volume of observations, enhanced utilization of DA, and improved background fields.

The assimilation of profiles has shown positive impacts on weather forecasting, particularly in enhancing short-term forecasts. This assimilation plays a crucial role in capturing

mesoscale weather phenomena, such as convective systems, thunderstorms, and localized rainfall patterns. It contributes to a more accurate representation of atmospheric processes, thereby improving the overall skill of weather forecasts, especially in regions with limited traditional observations [6]. Assimilating profiles provides a better understanding of the vertical structure of the atmosphere, facilitating accurate estimations of temperature, wind, and other variables' vertical distributions. Thus, it is of benefit for the analysis of atmospheric stability [7]. The real-time assimilation of profiles enables timely updates of atmospheric models, improving nowcasting and short-term forecasts [8]. This capability allows models to capture rapidly changing atmospheric conditions, providing crucial information for severe weather events and rapid weather developments [9].

Tropical cyclones exhibit increasing nonlinearity and dynamic instability in high-resolution models. Thus, it is essential to optimize the accuracy of forecasts through improved data assimilation. At the same time, the scarcity of real profile observations over the ocean results in significant challenges when evaluating assimilation results in oceanic regions. A comparison of the relative impacts of ocean-surface wind measurements and three-dimensional profiles on hurricane forecasts highlights the advantages of 3D wind measurements [10]. The different effects of assimilating only temperature, only winds, and both data types of temperature and wind observations in tropical regions concerning the background state in a perfect model have been explored [11]. However, due to cost limitations and practical constraints, the number of profile instruments cannot be infinite. Consequently, determining a cost-effective profile density that strikes a balance between construction costs and the error reduction benefits from assimilation effects becomes a crucial area of study. Simultaneously, the reduction in errors in data assimilation is influenced not only by profile density but also by the localization radius. In practical applications, a tuned localization radius of 1000 km is common for global modeling and data assimilation systems. However, for convective weather systems employing high-resolution models and observations, a much shorter radius of 10 km has been found to be more suitable [12]. Nevertheless, real-data experiments conducted by Dong et al. [13] indicated that a smaller localization radius is necessary to achieve a better analysis accuracy with denser observing networks. Periáñez et al. [14] derived an optimal localization radius through high-level heuristic arguments, assuming a uniform observing network, and they also recommend using a smaller localization radius for denser observations. These studies suggest a potentially intricate relationship between observing networks and localization radii.

To solve the consideration of both the observation cost and forecast benefit within data assimilation during the numerical forecast for mitigating typhoon disasters, this research aims to find a possibly resolved cost-effective profile density, that is an optimal density based on the relationship between the localization radius and the error reduction. The local error-subspace transform Kalman filter (LESTKF) [15] is applied for assimilating profile data into an atmospheric model named WRF-PDAF, which couples the WRF model [16] with a PDAF system (http://pdaf.awi.de, last accessed: 21 February 2023) [17]. The flowchart for this study is illustrated in Figure 1. This study conducts idealized case experiments to analyze assimilation results, with a primary focus on the online assimilation of profile data. In comparison to prior research, this study evaluates and contrasts the influence of assimilating profiles containing temperature (T), zonal wind (U), and meridional wind (V) at different densities on tropical cyclones. Additionally, twin experiments are performed with varied observation localization radii, aiming to identify the optimal localization radius for each observation density. Drawing on economic theories such as diminishing marginal utility [18] and increasing marginal costs [19], the study discusses and provides recommendations for achieving cost-effective profile density.

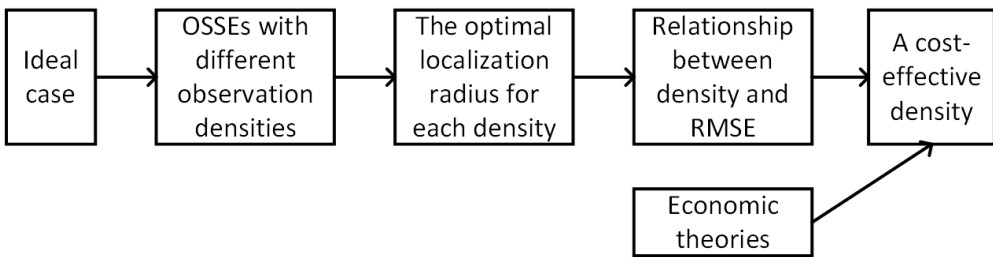

**Figure 1.** The flowchart of this study.

The subsequent sections of this study are structured as follows: Section 2 introduces the DA system setup and ensemble filters used for DA. Section 3 provides details of the experimental design for the idealized case studies. Section 4 presents the results, with a focus on profiles assimilated with different densities and selected localization radii. Section 5 offers a discussion and concludes the study.

## 2. Methodology

The main numerical model and assimilation method for solving the cost-effective profile density mainly includes WRF-PDAF, LESTKF, and the profile observation operator. The EnKF technique is a DA method that leverages an ensemble of model states and observations to update the model state variables. In this section, we introduce the WRF-PDAF model, LESTKF assimilation scheme, and the profile observation operators.

### 2.1. WRF-PDAF

The WRF model [16] is a widely used numerical weather prediction system known for its modular structure, allowing customization for specific research or operational forecasting objectives. The parallel data assimilation framework (PDAF) [17] offers a generic framework encompassing fully implemented and parallelized ensemble filter algorithms like LETKF, LESTKF, NETF, and LKNETF, along with related smoothers. PDAF facilitates model parallelization for parallel ensemble forecasts and includes routines for parallel communication between the model and filters. To introduce data assimilation capabilities, PDAF is integrated into the existing WRF framework.

WRF-PDAF, as developed by Shao and Nerger [20], couples WRF-ARW version 4.4.1 with PDAF to enable data assimilation. This coupling allows for the assimilation of profile data into WRF to enhance its initial conditions and, consequently, improve forecasts [21]. WRF-PDAF employs an online coupling strategy for data assimilation, utilizing a fully parallel structure. Sufficient processes are employed to run the data assimilation program concurrently with all ensemble states, ensuring each model task integrates only one model state, and the model consistently advances in time. This approach ensures high-efficiency data assimilation.

### 2.2. LESTKF

The LESTKF has found application in diverse studies involving the assimilation of satellite data into atmosphere, ocean, atmosphere–ocean coupled models, and hydrological models [22–25]. The LESTKF represents an efficient formulation of the EnKF and is introduced here to discuss the particularities of DA concerning the ensemble filter.

In the mathematical framework, each state vector is represented as $X^f$, transforming model fields into a one-dimensional vector. The analysis in Equation (1) transforms the forecast ensemble $X^f$, consisting of $N_e$ model states, into the analysis ensemble $X^a$:

$$X^a = X^f \left( w 1_{N_e}^T + \widetilde{W} \right) + \widetilde{x}^f 1_{N_e}^T \tag{1}$$

Here, $\widetilde{x}^f$ is the forecast ensemble mean state and $1_{N_e}^T$ is the transpose of a vector of size $N_e$ holding the value one in all elements. $w$ is a vector of size $N_e$, which transforms the ensemble mean, and $\widetilde{W}$ is a matrix of size $N_e \times N_e$, which transforms the ensemble perturbations:

$$w = TA(HX^fT)^T R^{-1}(y - H\widetilde{x}^f) \tag{2}$$

$$\widetilde{W} = \sqrt{N_e - 1} TA^{1/2}T^T \tag{3}$$

$A$ is a transform matrix in the error subspace.

$$A^{-1} = \alpha(N_e - 1)I + \left(HX^fT\right)^T R^{-1} HX^fT \tag{4}$$

$\alpha$ is the forgetting factor [26] used to inflate the ensemble to avoid underestimation of the forecast uncertainty. It leads to an inflation of the variance by $1/\alpha$. The forecast ensemble represents an error subspace of dimension $N_e - 1$, and the ensemble transformation matrix and vector are computed in this subspace. Practically, one computes an error-subspace matrix by $L = X^fT$, where $T$ is defined by Equation (5):

$$T_{j,i} = \begin{cases} 1 - \frac{1}{N_e} \frac{1}{\frac{1}{\sqrt{N_e}} + 1} & for\ i = j,\ j < N_e \\ -\frac{1}{N_e} \frac{1}{\frac{1}{\sqrt{N_e}} + 1} & for\ i \neq j,\ j < N_e \\ -\frac{1}{\sqrt{N_e}} & for\ i = N_e \end{cases} \tag{5}$$

$H$ is the observation operator. $R$ is the observation error covariance matrix. The matrix $A^{1/2}$ is computed using the eigenvalue decomposition of $A^{-1}$. $A^{1/2}$ and $A$ are computed using Equations (6)–(8):

$$USU^T = A^{-1} \tag{6}$$

$$A = US^{-1}U^T \tag{7}$$

$$A^{1/2} = US^{-1/2}U^T \tag{8}$$

where $U$ and $S$ denote the matrices of eigenvectors and eigenvalues.

The update on each grid point of the model is independent through a local analysis step. Observation localization is conducted based on horizontal and vertical influence radii when updating a grid point. Additionally, each observation is weighted according to its distance from the grid point [27], resulting in individual transformation weights $w$ and $\widetilde{W}$ for each local analysis domain.

### 2.3. Profile Observation Operators

Observation operators are employed to transform model variables into observation space, effectively computing the model equivalent of actual observations. In the context of profile data, the operators for temperature (T), zonal wind (U), and meridional wind (V) act directly on the model grid locations without any interpolations. Each profile comprises a vertical column of observations located on grid points, and each observation includes the three variables T, U, and V.

### 3. Experimental Design

Assimilation experiments considering both the observation cost indicated by the observation density and the model benefit indicated by the assimilation error reduction have been conducted, focusing on an ideal case. In this section, we outline the experimental design for conducting assimilation experiments with the WRF-PDAF model and describe the specifics of this ideal case, the twin experiments involving various profile densities and localization distances, and the objectives of these experiments.

### 3.1. Setup of the Twin Experiment

Tropical cyclones, known as hurricanes or typhoons, are formidable meteorological phenomena that originate over warm equatorial ocean waters. These powerful storms draw energy from the latent heat released as moist air rises and condenses into clouds and precipitation. Their rotation is influenced by the Coriolis effect, with the direction of spin determined by the hemisphere in which the cyclone forms. Given the severe impacts of tropical cyclones on coastal regions and infrastructure, accurate forecasting and monitoring are essential for mitigating their effects. For this study, we employ an idealized tropical cyclone case provided by the weather research and forecasting (WRF) model. This case serves as a simplified representation of real-world atmospheric conditions and offers a controlled environment for evaluating data assimilation methods through identical twin experiments.

The model domain spans 3000 km × 3000 km × 25 km, containing 200 × 200 × 20 grid points. The horizontal grid spacing is set at 15 km, and the vertical grid spacing is 1.25 km. The simulation covers a six-day period, starting from 1 September, at 00:00 UTC (010000), and concluding on 7 September, at 00:00 UTC (070000). The model time step is configured as 60 s. The initial state is characterized by motionlessness (u = v = 0) and horizontal homogeneity, except for the inclusion of an analytic axisymmetric vortex in hydrostatic and gradient-wind balance. Furthermore, periodic lateral boundary conditions are applied. The simulation employs the Kessler microphysics scheme and the YSU boundary-layer physics, with no utilization of radiation schemes. In the LESTKF method, an adaptive scheme [28] is adopted for the forgetting factor. Observation errors are assigned for temperature (T), horizontal wind (U), and vertical wind (V) at values of 1.2 K, 1.4 m/s, and 1.4 m/s, respectively, following Li et al. [11].

The true atmospheric state, encompassing temperature and wind fields, is derived from a forward run of the model and serves as the known reference for comparison with the assimilation results. This "truth" state is used to generate synthetic observations. A control state is separately generated from 3 September, at 12:00 UTC (031200), to 7 September, at 00:00 UTC (070000). This control state is generated with identical initial fields as those used for the true state. Thus, the control and true states are identical in all aspects except for their respective start times. The control simulation provides the initial state estimate for the subsequent data assimilation.

Synthetic observations are generated from the true state starting from 040800 and concluding at 051400, with hourly intervals. These observations include horizontal wind components (U and V) and temperature (T), and they are generated at all grid points within the model domain. Gaussian noise, with observation-error standard deviations, is added to all these synthetic observations.

In the twin experiments, an initial perturbation is introduced into the control state at 031200 to generate 40 ensemble members. This ensemble undergoes a 20 h spin-up period. Subsequently, observations are assimilated hourly into the ensemble during the analysis period from 040800 to 051400. Finally, an ensemble forecast is conducted without further assimilation from 051400 to 070000.

### 3.2. Experimental Design for the Cost-Effective Balance

The experimental design for data assimilation involves the incorporation of synthetic profile data into the WRF model. The impact of assimilating these observations on the model's representation of atmospheric variables, including temperature (T), horizontal wind (U), and vertical wind (V), is assessed through a comparison between the assimilated and true states. By conducting these experiments within an idealized setting, we aim to evaluate the performance and effectiveness of the WRF-PDAF system in assimilating observations and improving the model's representation of atmospheric variables. To achieve the most cost-effective balance, we need to determine the appropriate localization distance and profile density.

Two single runs are utilized to generate the true state (Experiment 4, 'True') and the control state (Experiment 5, 'CTRL'), as outlined in Section 3.1. These distinct states serve as the foundation for further analysis and experimentation in this study. To establish the initial ensemble, perturbations are introduced to the initial control state using second-order exact sampling [26] based on the model variability observed in hourly snapshots from 010000 to 031200. The central state is set to the control state at 031200. Subsequently, a free ensemble run involving 40 members (Experiment 6, 'ENS') is conducted. The purpose of this ensemble run is to generate a collection of model states encompassing a range of potential variations and uncertainties. The same initial ensemble members are employed in the assimilation experiments. Building on the results of the ensemble run, assimilation experiments are carried out using profiles from 30 analysis cycles.

In this study, we consider the localization radius and profile densities as key factors for determining the most cost-effective profile density. Various localization radii are examined, including $0dx$, $3dx$, $5dx$, $10dx$, $20dx$, and $30dx$, where $dx$ represents the grid distance in the $x$-direction (set at 15 km), with the grid distance in the $y$-direction also equal to $dx$. The profile density is determined based on the placement of observations. When observations are located on all grid points, the density is defined as 100%. Consequently, when observations are placed at every 2 grid points in both the $x$ and $y$ directions, the density is 25%. Observations at every 3 grid points in both directions result in a density of 1/9, approximately 11.1%. Similarly, observations at every 5 grid points in both directions yield a density of 4%, while observations at every 10 grid points in both directions correspond to a density of 1%. Assimilation experiments are conducted using profiles with different densities, including 100%, 25%, 11.1%, 4%, and 1%. Various experiments assimilating U, V, and T variables with different localization radii and densities, as listed in Table 1, are performed to evaluate the impact of assimilating observations from profiles with different densities on the model state. The aim of these tests is to select the most suitable profile density that achieves the most cost-effective balance.

**Table 1.** The design for localization radii and profile densities.

| Exp | Name | Member(s) | Profile Density (%) | Localization (km) | DA-Cycle (s) |
|-----|------|-----------|---------------------|-------------------|--------------|
| 1 | True | 1 | - | - | - |
| 2 | CTRL | 1 | - | - | - |
| 3 | ENS | 40 | - | - | - |
| 4 | D100L0 | 40 | 100 | $0dx$ | 30 |
| 5 | D100L3 | 40 | 100 | $3dx$ | 30 |
| 6 | D100L5 | 40 | 100 | $5dx$ | 30 |
| 7 | D100L10 | 40 | 100 | $10dx$ | 30 |
| 8 | D25L0 | 40 | 25 | $0dx$ | 30 |
| 9 | D25L3 | 40 | 25 | $3dx$ | 30 |
| 10 | D25L5 | 40 | 25 | $5dx$ | 30 |
| 11 | D25L10 | 40 | 25 | $10dx$ | 30 |
| 12 | D11L0 | 40 | 11.1 | $0dx$ | 30 |
| 13 | D11L5 | 40 | 11.1 | $5dx$ | 30 |
| 14 | D11L10 | 40 | 11.1 | $10dx$ | 30 |
| 15 | D11L20 | 40 | 11.1 | $20dx$ | 30 |
| 16 | D4L0 | 40 | 4 | $0dx$ | 30 |
| 17 | D4L5 | 40 | 4 | $5dx$ | 30 |
| 18 | D4L10 | 40 | 4 | $10dx$ | 30 |
| 19 | D4L20 | 40 | 4 | $20dx$ | 30 |
| 20 | D1L0 | 40 | 1 | $0dx$ | 30 |
| 21 | D1L5 | 40 | 1 | $5dx$ | 30 |
| 22 | D1L10 | 40 | 1 | $10dx$ | 30 |
| 23 | D1L20 | 40 | 1 | $20dx$ | 30 |
| 24 | D1L30 | 40 | 1 | $30dx$ | 30 |

## 4. Results and Analysis

### 4.1. Relationship between Localization Radii and Observation Densities

Figure 2 shows the RMSEs of U and V from 031200 to 070000 using different observation densities and localization radii. With fixed observation errors, a sparser observing network (the density less than 4%) favors a larger localization radius to achieve the best filter performance, and a denser network favors a smaller localization radius, which is consistent with findings from the study of Ying et al. [29]. It is worth noting that the optimal localization radii for the different densities are not the smallest when the density is greater than 11.1%. Furthermore, in time series, the RMSEs reached the minimum value at 051400, and no significant improvement was observed thereafter during the analysis period. Therefore, RMSEs at 051400 are also selected to find the optimal localization radii.

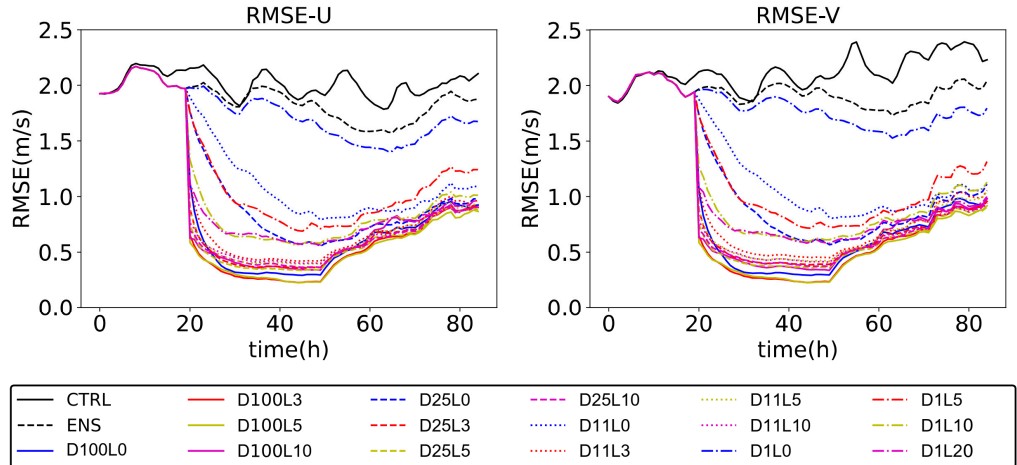

**Figure 2.** The RMSEs of U and V from 031200 to 070000 using different observation densities and localization radii (RMSE of U: m/s, RMSE of V: m/s).

In our study, we firstly focus on determining the optimal localization radius in data assimilation (DA) for different observation densities, as this plays a critical role in DA effectiveness. We use the root-mean-square error (RMSE) as a key metric to evaluate the performance of different combinations of localization radii and observation densities. Table 2 shows the RMSEs for the temperature (T) and horizontal wind components (U and V) at a specific time (051400) for various experiments listed in Table 1. The RMSEs for the ensemble forecast (ENS) are consistently lower than those for the control run (CTRL) based on the true state (True). This suggests that the ensemble approach itself improves the accuracy of the model prediction, and the ensemble mean provides an accurate forecast. When assimilating U, V, and T data, the RMSEs are generally lower than those of ENS.

**Table 2.** The RMSEs of T, U, and V for experiments in Table 1 at 051400. **The numbers in bold represent the smallest values in each density group**.

| Exp | Name | RMSE_T (K) | RMSE_U (m/s) | RMSE_V (m/s) |
|-----|--------|------------|--------------|--------------|
| 1 | True | - | - | - |
| 2 | CTRL | 1.112 | 1.929 | 2.063 |
| 3 | ENS | 0.939 | 1.799 | 1.910 |
| 4 | D100L0 | 0.185 | 0.294 | 0.294 |
| 5 | D100L3 | **0.145** | 0.233 | 0.234 |
| 6 | D100L5 | 0.148 | **0.229** | **0.230** |
| 7 | D100L10 | 0.251 | 0.337 | 0.337 |
| 8 | D25L0 | 0.430 | 0.553 | 0.567 |
| 9 | D25L3 | 0.249 | 0.361 | 0.389 |
| 10 | D25L5 | **0.239** | **0.339** | **0.361** |

**Table 2.** *Cont.*

| Exp | Name | RMSE_T (K) | RMSE_U (m/s) | RMSE_V (m/s) |
|---|---|---|---|---|
| 11 | D25L10 | 0.273 | 0.361 | 0.371 |
| 12 | D11L0 | 0.583 | 0.790 | 0.811 |
| 13 | D11L3 | 0.310 | 0.418 | 0.450 |
| 14 | D11L5 | **0.285** | **0.393** | 0.418 |
| 15 | D11L10 | 0.300 | 0.396 | **0.409** |
| 16 | D4L0 | 0.765 | 1.21 | 1.26 |
| 17 | D4L5 | 0.381 | 0.468 | 0.501 |
| 18 | D4L10 | **0.353** | **0.454** | **0.471** |
| 19 | D4L20 | 0.405 | 0.513 | 0.522 |
| 20 | D1L0 | 0.895 | 1.617 | 1.709 |
| 21 | D1L5 | 0.673 | 0.710 | 0.727 |
| 22 | D1L10 | 0.566 | 0.580 | 0.611 |
| 23 | D1L20 | **0.481** | **0.563** | **0.579** |
| 24 | D1L30 | 0.486 | 0.666 | 0.677 |

Figure 3 displays the values listed in Table 2. These experiments include the control run (CTRL), ensemble forecast (ENS), and different DA experiments with varying localization radii (D) and observation densities (L). This figure shows the relationship between assimilating different observation densities and RMSE in the case of variable influence radius. This figure proves that the value we selected from Table 2 is indeed the smallest.

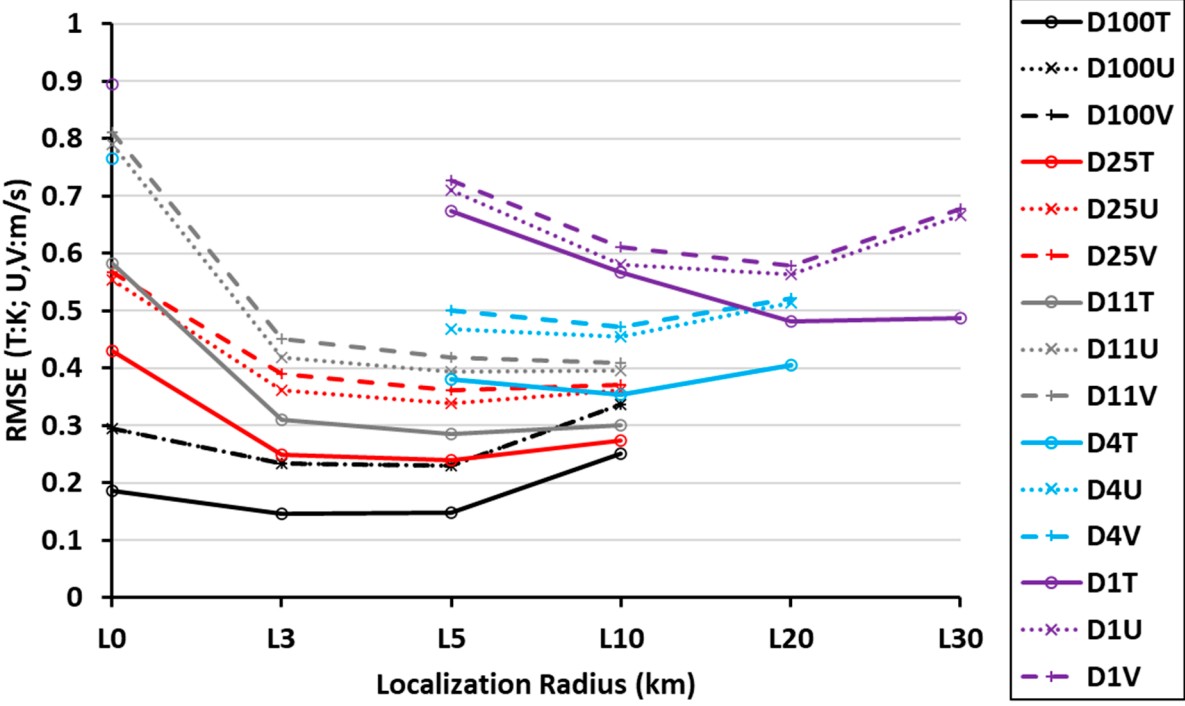

**Figure 3.** The RMSEs of T, U, and V at 051400 using different localization radii (T: K, U: m/s, V: m/s).

### 4.2. The Most Cost-Effective Balance

The lowest RMSEs are selected from Table 2 for each combination of observation densities. Then, the relationships between RMSEs for T, U, V, and observation densities are shown in Figure 4. These relationships are characterized by high correlation coefficients, as indicated by all the coefficients of determination ($R^2$) exceeding 0.99.

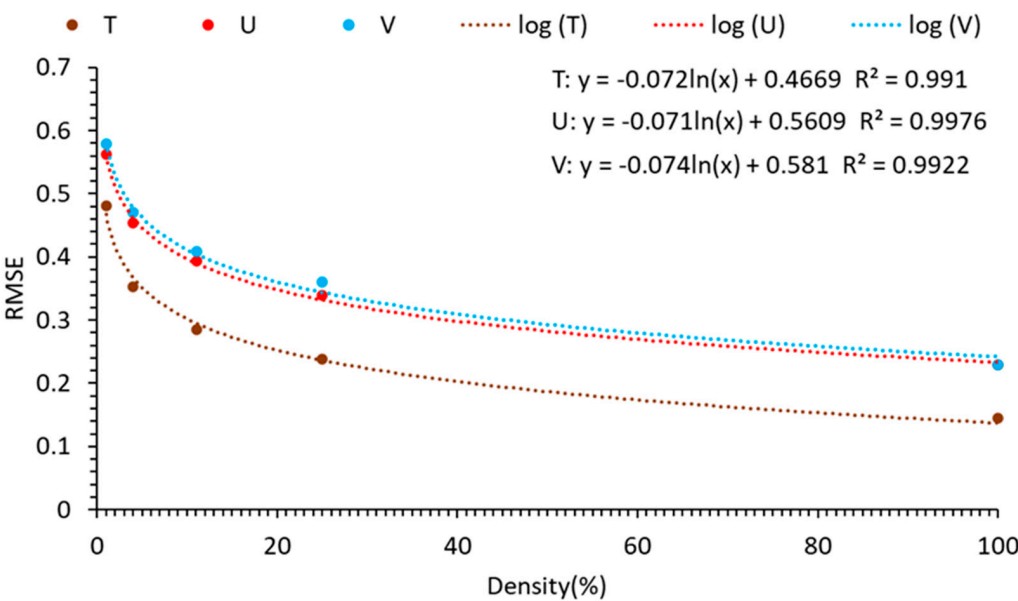

**Figure 4.** The variation in RMSEs with density (T: K, U: m/s, V: m/s).

To identify the most cost-effective balance between observation density ($x$) and benefit/cost, we introduce two essential functions: the benefit function $f(x)$ and the cost function $g(x)$. Here are two key assumptions and steps for finding the most cost-effective balance:

- The cost is defined as the deployment of observations. The cost function is defined as the linear relationship between the cost and density, i.e., $g(x) = x$. Thus, the total cost of fully deploying observations at 100% density is considered 100%, and no cost if no deployment (0 density).
- The benefit is defined as the property saved due to the reduction in the RMSE. Here, the total property can be saved is defined as $a$. The relationship between the property saved and the RMSE reduction in wind is linear [30,31]. The RMSE–density relationship follows Figure 4 and is denoted as $r(x)$. Therefore, the relationship between the benefit and density is $f(x) = a - r(x)$, representing the benefit function. Thereby, when density is 100%, the benefit is $a$. Conversely, when density is 0, the benefit is 0.

Based on these assumptions and economic theories like diminishing marginal utility [18] and increasing marginal costs [19], the most cost-effective balance is reached when the reduction rate of the benefit function $f'(x)$ equals 1. In this context, the optimal observation densities for U and V are found to be 26.6% and 27.2%, respectively. When the observation density is lower than this balance density, the rate of benefit increment exceeds the rate of cost increment, resulting in significant benefits and willingness for construction. Conversely, when the density is higher than the balance density, the rate of benefit increment is smaller than the rate of cost increment, leading to smaller benefits and willingness for construction. This analysis helps to identify the observation densities that provide the most cost-effective balance for various atmospheric variables in the context of data assimilation.

## 5. Discussion and Conclusions

The study presented herein delves into the realm of data assimilation (DA) within meteorology, with a particular focus on the impact of observation density and localization radius. Several noteworthy points emerge from this investigation.

- Significance of Profile DA

This study reaffirms the pivotal role of profile DA techniques in enhancing the precision of meteorological models. The meticulous assimilation of profile data into numerical

models is imperative for the accurate prediction of weather phenomena, especially complex events like tropical cyclones.

- Influence of Observation Density and Localization Radius

The research extensively examines the influence of two critical parameters: observation density and localization radius. The investigation reveals that these parameters are intrinsically intertwined and significantly affect the accuracy of DA outcomes. Especially when the density is small, a larger localization radius is essential. The empirical results suggest that a careful balance must be struck between the two.

Observation Density: The study demonstrates that observation density plays a pivotal role in the efficacy of DA. It highlights that excessively sparse observations can lead to a reduced accuracy in model predictions. Conversely, excessive observation density may not yield commensurate improvements and could incur unwarranted costs. The findings indicate that a sweet spot exists, typically around 26–27% observation density for the U and V variables in the context of this study. Beyond this optimal point, the incremental benefits in forecast accuracy diminish relative to the cost of deploying additional instruments.

Localization Radius: Equally crucial is the localization radius, which determines the spatial influence of observations on the model. The research emphasizes that the localization radius must be carefully tailored to the observation density. For instance, when the observation density is relatively low, a larger localization radius might be necessary to effectively incorporate sparse data into the model. Conversely, higher observation densities may require smaller localization radii to avoid overfitting.

- Practical Implications

These findings have practical implications for meteorological data collection efforts. It underscores the importance of strategically deploying observation instruments. An optimal balance between cost-effectiveness and forecast accuracy must be struck. Moreover, the study encourages meteorological agencies to consider not only the quantity of observations but also their spatial distribution and the appropriate scale of influence.

Figure 5 shows the wind distributions of the TC at 051400 on the surface level (10 m wind), which could have possibly the greatest impact on the layout of observation equipment as they usually cause accountable disasters during its application process. It is clear that the surface wind vectors from ENS and 11.1% density are denser than the ideal truth, but miss the spatial details and also have a weak strength. This indicates that the spatial layout density or cost can somewhat greatly affect the numerical forecast or benefit. Moreover, surface wind vectors from 26 to 27% and the 100% observation density are almost equitable to the ideal truth (not shown). This indicates that almost 1/4 of the numerical gridded resolution could be the possible best solution for the ground-based profile observation layout when considering mitigating the disasters caused by typhoon winds. It is worth noting that the ensemble mean fields in Figure 5 are simply averaged over ensemble members, disregarding the TC structures in individual ensemble members. As an alternative, the ensemble mean fields can be calculated based on the feature-oriented mean (FM) method, to avoid the unrealistic smoothing of the TC structure and an underestimation of the intensity [32].

In conclusion, this study sheds light on the critical interplay between observation density and localization radius in data assimilation. The research underscores that the pursuit of higher observation densities should be tempered by careful consideration of the localization radius. The identified sweet spot of around 26–27% observation density for U and V variables, coupled with appropriately matched localization radii, is indicative of a cost-effective balance that maximizes the utility from forecast accuracy. These insights are invaluable for meteorological agencies and researchers, guiding them in making informed decisions about observation network designs and improving the precision of weather predictions. In the future, we will deploy real profile observations partly according to this study. Then, we can carry out observation system experiments based on the real TC cases. In addition, target observation, i.e., the approach of conditional nonlinear optimal

perturbation (CNOP), proposed by Mu et al. [33], will be considered, and may provide better recommendations for observation network designs. The CNOP has been applied to the studies of target observations associated with TCs, which have achieved positive effects on TC forecasts [34,35].

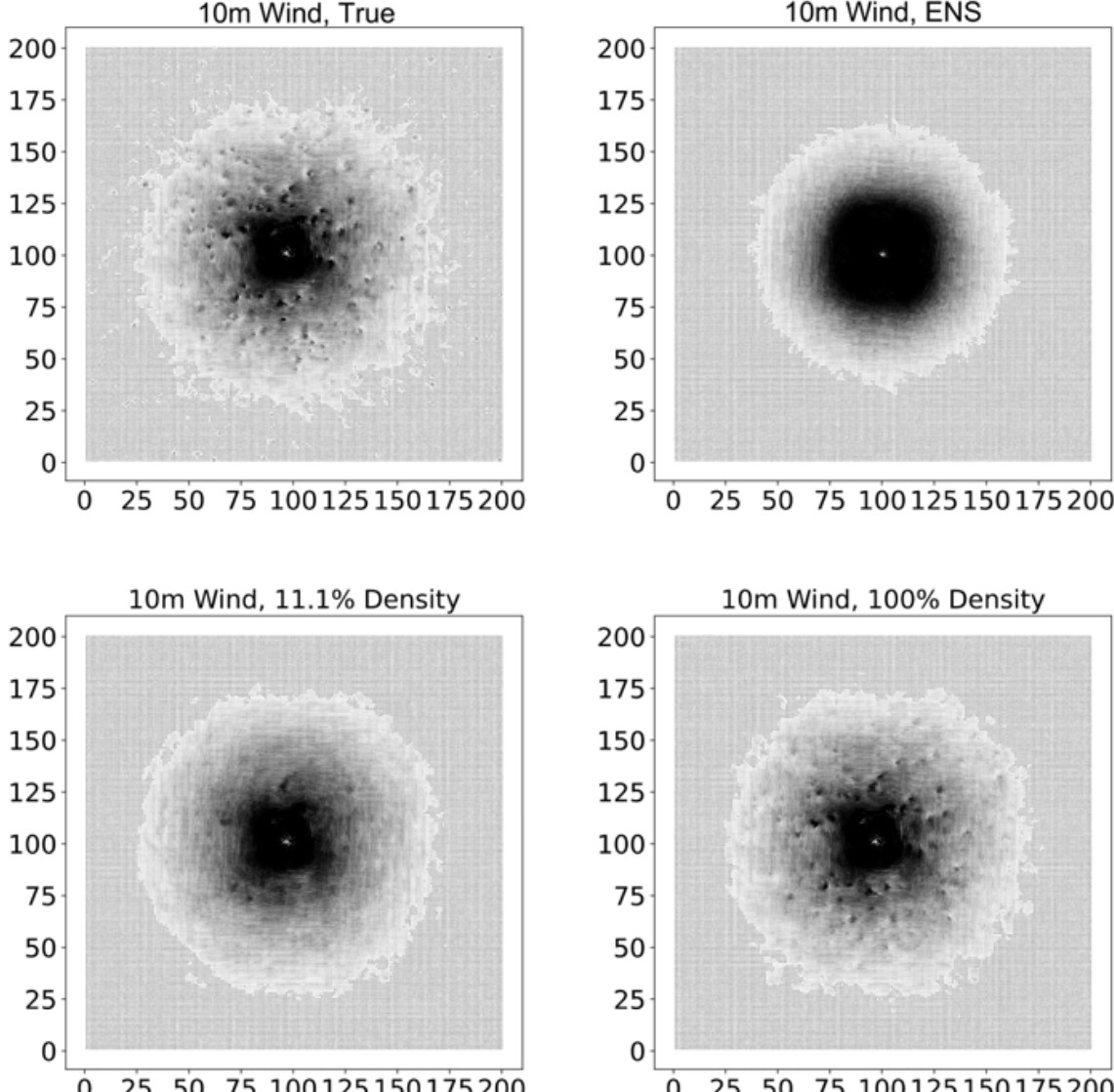

**Figure 5.** The distributions of TC at 051400 on the surface level (10 m wind: m/s).

**Author Contributions:** Conceptualization, C.S.; methodology, C.S.; software, C.S.; validation, C.S.; formal analysis, C.S.; investigation, C.S.; resources, L.N.; data curation, C.S.; writing—original draft preparation, C.S.; writing—review and editing, L.N.; visualization, C.S.; supervision, L.N.; project administration, C.S.; funding acquisition, C.S. All authors have read and agreed to the published version of the manuscript.

**Funding:** Changliang Shao (No. 202105330044) is supported by the China Scholarship Council for one year's research at AWI. This work is supported by the National Natural Science Foundation of China (Grant No. 41705133), Key Laboratory of Space Ocean Remote Sensing and Application.

**Data Availability Statement:** Dataset can be download at https://doi.org/10.5281/zenodo.10254615, accessed on 5 November 2023.

**Conflicts of Interest:** The authors declare no conflicts of interest.

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
