# Peer review of "The Impact of Profiles Data Assimilation on an Ideal Tropical Cyclone Case"

_remotesensing, doi:10.3390/rs16020430_

Round 1
Reviewer 1 Report
Comments and Suggestions for Authors
This manuscript was going to examine the impact of assimilating profiles to simulate tropical cyclone. The topics including the impact of profiles, density of observations and localization radii, and cost-effective of the observations were discussed. For the results of the study, there is a major issue should be carefully considered again.
The main topic of this study is not clear, the issues of impact of profiles, the parameters in data assimilation method, and design of the observations network were touched without detailed discussions. For example, the title of the study is the impact of profiles with a conclusion is that profile is important. It is not interesting. It is well known that profiles are important and helpful for tropical cyclone simulation. Is there any new finding? There are a few sentences related to the cost-effective balance between observation density and benefit without detailed research. The relationship between density of observation and localization radius maybe only exists in this idealized experiment. For the real circumstance, it is difficult or impossible to determine the localization radius according to the density of the observation. Practically, the localization radius is not determined by density of the observation network. Authors should work on one issue and try to give some new findings.
Reviewer 2 Report
Comments and Suggestions for Authors
Review of “The Impact of Profiles Data Assimilation on an Ideal Tropical 2
Cyclone Case” by Shao et al.
This study evaluates the influence of profile data assimilation (DA) configurations in enhancing weather predictions though an ideal tropical cyclone (TC) case. The results provide a useful information to balance the cost-effectiveness and forecast accuracy. The manuscript is logically organized and well written. However, I think some major points need to be addressed and clarified before the publication of this manuscript. The following comments are some concerns and suggestions.
1. The prediction of TC track and intensity has always drawn significant attention due to its importance in TC disaster preparedness and prevention. In this manuscript, the author only consider the improvement of atmospheric variables from DA, but does not examine whether this can lead to benefit for TC track and intensity forecasts. I suggest the author analyze the effect of DA on the spatial distribution of the wind and temperature field. If the DA improves the environmental steering flow, the track errors may be reduced. Better description of TC structure is benefit of intensity forecasts.
2. In lines 176-181, they are not clearly presented and need to give more detail.
3. In lines 263-274, the author gives two key assumptions for finding the best cost-effective balance. Among them, there are two important linear relationships. One is between the cost and density, while another is between the property save and the RMSE reduction of wind. Why the author give these relationships? Is it common?
4. The Fig. 3 mentioned in line 271 cannot be found in this manuscript.
5. The figures and tables in this manuscript only show the RMSEs at 051400. How about the performance of forecasts without further assimilation from 051400 until 070000? Does the author evaluates potential configurations for future profile measurements only based on the RMSEs at 051400. If so, why?
6. The manuscript give a useful method to identify the cost-effective profile density. The results show that the optimal observation densities for U and V are around 26-27% (line 306), which are computed through an idealized TC case. However, the optimal densities may be very different for TCs with different sizes and intensities. If so, what is the optimal observation density provided to the meteorological agencies and researchers? I suggest the author can conduct experiments foe more TC cases with different characteristics to see the range of the optimal observation densities.
Comments on the Quality of English LanguageThe writing should be improved.
Round 2
Reviewer 2 Report
Comments and Suggestions for Authors
Review of “The Impact of Profiles Data Assimilation on an Ideal Tropical 2
Cyclone Case” by Shao et al.
The authors have made some improvements to the manuscript and have addressed my concerns to a large degree. Thanks to the authors for their thoroughness in their responses. I just have a few additional minor suggestions.
1. In Line 397-399, the author says that the tropical cyclone (TC) in the ENS has a weak strength than that in the ideal truth. Here, the intensity in the ENS appears to be derived from the ensemble mean field in Figure 5. Is the ensemble mean field simply averaged over all ensemble members, disregarding the TC structures in individual ensemble members? The ensemble mean can filter out some unpredictable features in the forecast member fields and reduce the forecast errors relative to deterministic forecasts. But for TC, such averaging is well-known to cause unrealistic smoothing of the TC structure and an underestimation of the intensity (Zhang et al., 2021). As a consequence, the TC intensity usually cannot directly derive from the ensemble mean field. Thus, the results in line 399 needs further justification. To clarify that, the author can calculate the ensemble mean forecast in Figure 5 based on the feature-oriented mean (FM) method in Zhang et al. (2021). Or the TC intensity forecasts of ensemble mean, more generally, can be represented by the ensemble-averaged MWS for all members. The MSW for each member represent the maximum 10 m wind speed in the TC inner core region, which were obtained from the model grids.
Zhang, J., J. Feng, H. Li, Y. Zhu, X. Zhi, and F. Zhang, 2021: Unified Ensemble Mean Forecasting of Tropical Cyclones Based on the Feature-Oriented Mean Method. Wea. Forecasting, 36, 1945–1959, https://doi.org/10.1175/WAF-D-21-0062.1.
2. In this study, the assimilation strategy was to assimilate observations on evenly distributed regular grids over TC. However, there may exist some key areas over TC where additional observations are expected to have a large contribution to reducing the TC forecast errors. Thus, deploy more profile observations in these areas can maximize the utility from forecast accuracy. Such observational strategy is generally called “target observation”. Mu et al. (2003) proposed a nonlinear technique of target observation, i.e., the approach of conditional nonlinear optimal perturbation (CNOP). The CNOP has been applied to the studies of target observations associated with TCs and positive effects on TC forecasts are achieved (Mu et al., 2009; Qin et al., 2022). Given the forecast benefit within data assimilation, combine the method in this study with the CNOP method may help provide better recommendations for observation network designs. The author can consider this in the future.
Mu, M., W. S. Duan, and B. Wang, 2003: Conditional nonlinear optimal perturbation and its applications. Nonlinear Processes in Geophysics, 10, 493-501, https://doi.org/10.5194/npg-10-493-2003
Mu, M., F. F. Zhou, and H. L. Wang, 2009: A method for identifying the sensitive areas in targeted observations for tropical cyclone prediction: Conditional Nonlinear Optimal Perturbation. Monthly Weather Review, 137, 1623-1639, https://doi.org/10.1175/2008mwr2640.1
Qin, X. H., W. S. Duan, P.-W. Chan, B. Y. Chen, and K.-N. Huang, 2022: Effects of dropsonde data in field campaigns on forecasts of tropical cyclones over the Western North Pacific in 2020 and the role of CNOP sensitivity. Advances in Atmospheric Sciences, https://doi.org/10.1007/s00376-022-2136-9
Comments on the Quality of English Language
N/A
